# I DiG STEM: A Teacher Professional Development on Equitable Digital Game-Based Learning

**Anthony Muro Villa III [1],\*, Quentin C. Sedlacek [2] and Holly Yvonne Pope [3]**

1   School of Education, University of California, Riverside, 900 University Ave., Riverside, CA 92521, USA
2   Simmons School of Education & Human Development, Southern Methodist University, 6401 Airline Rd, Suite 301, Dallas, TX 75205, USA; qsedlacek@smu.edu
3   Allegheny Intermediate Unit, Homestead, PA 15120, USA; holly.pope@aiu3.net
*   Correspondence: avilla@ucr.edu

**Abstract:** Digital game-based learning (DGBL) has the potential to promote equity in K–12 STEM education. However, few teachers have expertise in DBGL, and few professional development models exist to support teachers in both acquiring this expertise and advancing equity. To support the development of such models, we conducted a professional development to explore teacher acquisition of technological, pedagogical, and content knowledge for games (TPACK-G) during a DGBL workshop series informed by culturally relevant pedagogy. This mixed methods pilot study used pre- and post-surveys and interviews to investigate shifts in teachers' (*n* = 9) TPACK-G, perceptions of DGBL, and operationalizations of equity and cultural relevance. The survey findings showed increases in teachers' TPACK-G, and corroboration between the surveys and interviews showed teachers' expanded ideas about the range of applications of digital games in STEM education. However, the interviews revealed that teachers' conceptualizations of equity and cultural relevance varied considerably.

**Keywords:** teacher professional development; digital game-based learning; Digital Game Evaluation Framework; pedagogical and content knowledge; TPACK-G; equity; cultural relevance; STEM

## 1. Introduction

Science, technology, engineering, and mathematics (STEM) education is a crucial foundation for increasing students' opportunities and skills. Yet, systemic racism, lack of teacher preparation, and other forms of inequities deprive many students of access to high-quality STEM learning [1–4]. Digital game-based learning (DGBL) may be one useful tool for disrupting these inequities. Digital games can be implemented in a variety of contexts and can be intrinsically interesting to students [5]. Under certain conditions, DGBL has been shown to increase motivation, engagement, and deep learning [6,7]. DGBL can also promote twenty-first-century skills such as adaptation, self-monitoring, and problem solving, and can prepare students for future learning [8–10].

However, the fact that digital games can be effective does not mean they inherently are. Teachers who search the Internet for digital games will discover an overwhelming abundance of purportedly 'educational' games of varying quality, and these same teachers often receive little or no support in choosing and using games to promote equitable learning. Hu and Sperling [11] found that many preservice teachers struggled to recognize differences in the educational value of different games and had narrow conceptions of the instructional utility of games in general. These obstacles have complex roots. For example, many digital games with mathematics or science content are targeted towards elementary-age children, and elementary teachers often experience anxiety about their ability to teach mathematics and science content [12]. This is due, in part, to the relatively few mathematics and science courses in teacher preparation programs for preservice elementary teachers [13–15]. Also, US educational policies since the No Child Left Behind Act of 2002 have often led

to tradeoffs, shifting funds to improve standardized testing scores in mathematics and language arts while sometimes reducing focus on science [16–18].

Not only do teachers need more content knowledge but they also require pedagogical and technological knowledge that can enable them to teach STEM content effectively. Specifically, teachers need more development in technological pedagogical content knowledge for games (TPACK-G). As a field, we must prepare teachers to "choose and use" digital games effectively and support teachers in critically analyzing how digital games can advance equity or (re)produce inequity. With these goals in mind, the present study presents the design and results of a pilot study of professional development (PD) called Implementing Digital Games for STEM (I DiG STEM) focused on applying DGBL for STEM lessons to advance educational equity. This study was guided by the following research questions: (1) How did the PD increase teachers' TPACK-G? (2) How did teachers' perceptions and implementation of DGBL shift? (3) How did teachers operationalize concepts of equity and cultural relevance? This study used mixed methods to answer these research questions. We conducted pre- and post-surveys to measure changes in teachers' TPACK-G and perceptions of DGBL. To gain additional insight into how teachers' perceptions and implementation of DGBL shifted, we interviewed seven of the nine participants. The interview also served to investigate how teachers were operationalizing concepts of equity and cultural relevance after implementing a digital game-based lesson.

### 1.1. Digital Game-Based Learning (DGBL)

DGBL refers to a teaching method that utilizes digital games to engage students in learning or applying educational content. Research suggests that games can be used for sustained, experiential learning of educational content [7,19], and that games can motivate students and engage them in critical thinking and problem solving [20]. Additionally, DGBL has sometimes been found to improve a specific subset of cognitive skills called executive function, which are required to plan, monitor, and achieve goals [21,22]. In a STEM-specific context, DGBL can engage students in deep learning and promote the development of STEM skills [23–25], and has sometimes shown positive correlations with persistence in STEM [26,27]. While the term DGBL encompasses innumerable experiences of differing quality, it certainly encompasses many experiences with the potential to be extremely valuable for STEM education.

It is far from clear that this potential is being widely realized. Although technology is more accessible than ever to teachers and students in many schools, research reveals that many teachers are only implementing technology as a substitute for activities that could be completed without technology [28,29]. Teachers report even less integration of technology when serving low-income or urban students [30–32]. Teachers note challenges to incorporating technology into their curriculum that include pressure to teach to standardized tests, lack of resources or access to digital technologies, and lack of technical support [33,34]. Teachers further express that PD alone is not enough to integrate technology into the curriculum; they cite the need for more knowledge, skills, resources, and support [35]. With these needs in mind, we turn to describing a framework that can support teachers' selection and implementation of digital games.

### 1.2. Defining Equity

Equity can defined as the fair distribution of opportunities to learn, equal opportunites to participate, and equal educational outcomes [36,37]. The fair distribution of opportunities to learn has long been framed by socological approaches generally around the resources available for a student to achieve (e.g., textbooks, instructional strategies, materials, teacher's background, class size, time, and quality) [38–40]. Tate [41] extends this definition to emphasize the importance technology has for learning science.

*1.3. Digital Game Evaluation Framework*

Not all digital games contribute to learning in the same ways or to the same extent. In conducting an Internet search for digital games that could support STEM learning, we found that the quality of games and the level of thinking required to play them varied greatly [42]. Many games use low-level cognitive processes, such as basic recall and procedures [43]. Some games mimic digital worksheets where gameplay is wholly unrelated to STEM content; in these, gameplay is periodically interrupted with prompts that ask students to solve mathematics problems or choose correct answers to science questions. Notably, providing students with closed-ended tasks such as these may influence students' development of fixed mindsets about the discipline [44]. On the other hand, some games show much greater potential to support higher-level cognitive processes [6]. Educators need to be able to differentiate between games that support higher- and lower-level cognitive processes.

The benefits of digital games (or lack thereof) may also be shaped by questions of identity and representation. For example, studies have shown that a lack of diverse representation in games can have negative effects on players of color and women [45–49]. Games may send students more implicit messages about culture and identity as well [50]. Relatedly, STEM identity is influenced by cultural models and messages around who is "good" at mathematics [51,52]. Thus, when selecting digital tools for learning, both explicit and implicit messages that are sent through the interface of the game must also be addressed.

These concerns point to the need for a clear framework to help educators select games that would contribute to equitable STEM learning experiences. Previous frameworks developed for choosing and using digital games have focused on identifying age-appropriate games that fit the technical and logistical constraints of the classroom environment [53]. A new framework is needed that addresses learning potential by considering the cognitive and sociocultural dimensions of student learning. Author3 developed such a framework to help teachers critically analyze the types of games and technology we bring into the classroom. This framework was developed and used in previous studies to analyze digital games used in classrooms (see [42,52]). It consists of three main components: game features, learning complexity, and cultural relevance. The framework consists of guiding questions to elicit analysis based on the following conceptualizations of each component (see Figure 1).

| Component | Example Questions to Ask |
|---|---|
| **Game Features** | • Is content (and learning mechanic) incorporated into the game mechanics, or is it treated as a separate add-on? <br> • Do the artistic features of the game support engagement in learning? <br> • How does the game involve speed or time? |
| **Learning Complexity** | • Does the game require players to engage in high or low levels of thinking? <br> • Are there multiple entry and exit points? <br> • Are there many ways to solve or approach mastery at each level? |
| **Cultural Relevance** | • Does the game's backstory or theme send cultural or gendered messages that may negatively affect student learning? <br> • Does the game send fixed mindset messages by its treatment of mistakes? <br> • Does the game contain options for personal choice or preference? |

**Figure 1.** Digital Game Evaluation Framework.

### 1.3.1. Game Features

Borrowing from the field of game design, we define game features as artistic features, game mechanics, and related aspects of gameplay design [54]. The most engaging games, by design, connect the player to the game through its features, allowing the player to master (or learn) the game as they progress through the levels [9,55]. The artistic features (e.g., graphics, animations, etc.) may influence how a player becomes interested and engaged in the game, which is important for learning [54]. Game mechanics describe how players are expected to interact with the rules, goals of play, player actions and strategies, and concrete game traits [56–58]. Unfortunately, many educational games are artistically appealing but use content that is not directly embedded in the gameplay. Educators should consider how the learning mechanics of academic content are integrated with the game mechanics.

Learning mechanics are a model proposed by Lim and others [59] to guide serious game design or game analysis by directly linking elements of gameplay to the pedagogical intent of the game. Arnab and others [60] extended this model to consider the concrete and abstract elements of learning potential of a game. When games separate learning content from gameplay, this is known as 'gamification' of academic tasks [61]. For example, a game may require players to solve a series of multiplication problems, and then—after correctly completing a certain number of these problems—the game may require players to navigate an obstacle course wholly unrelated to multiplication. Research suggests this is not the most effective or engaging design for an educational game; games that integrate the game mechanic with the content may be more effective and engaging. As Habgood and Ainsworth [62] found, students who played a mathematics game with integrated content and game features learned more than students who played the same game but with the content as a separate add-on component. For example, a game that integrates learning content with game mechanics might invite students to learn multiplication skills by making players' manipulation of area models the primary mechanism of gameplay.

The speed and time requirements of games can also create barriers to gameplay. For educational games, speed and limited time to engage with content can contribute to false, overly narrow perceptions about ability, particularly in mathematics [63,64].

The game features component of the framework asks teachers to consider how content is integrated into the game mechanics, how the aesthetics of the game work with that content, and whether time limits or speed requirements are part of the game. Although many other game features may play a role in mediating student learning, these three provide a starting point to help teachers begin critically analyzing the affordances and limitations of games for supporting STEM learning.

### 1.3.2. Learning Complexity

Along with analyzing how content is (or is not) embedded within gameplay, it is also important to consider the potential learning that can come from engaging with the game. Learning complexity interrogates the prospect of digital games for supporting deep learning processes. Here, we draw upon the framework of cognitive demand, which categorizes learning activities by their potential (and implementation for) critical thinking and cognitive processes while solving a task [65,66]. The learning complexity of a game is greatly influenced by its game features but also how a teacher chooses to implement it, as well as the relationship of game tasks to students' prior experiences [67–69].

One aspect of a digital game's cognitive demand is intrinsic integration, as discussed previously. Games where the content is separated from the play tend to have relatively low levels of cognitive demand, asking players to either recall information (memorization) or enact procedures without necessarily connecting these procedures to conceptual understanding or applications (procedures without connections) (cf. [65,70]). For games, this means that play is interrupted with activities where students are repeating information, solving with rules or algorithms, or memorizing facts [71]. For example, some educational games are simply digitized worksheets [72]. These types of games are considered "unidimensional" and do not promote deep learning [73]. On the other hand, some games may

place high-level cognitive demands upon students, pushing them to engage with mathematics or science through more conceptual, immersive, and meaning-making activities [66,69]. Digital games that involve students in high levels of cognitive demand as an integral part of gameplay contribute to deep learning and engagement [55]. Games with a high level of learning complexity tend to be those where the content is intrinsically integrated.

One example of a game with high cognitive demand is the immersive *Tyto Online* (tytoonline.com), which we used to introduce participants to DGBL during our PD. The game provides players a variety of scientific problems or "storylines" to explore and solve. For example, in one storyline, players take on the role of an investigative scientist helping to care for a monk seal with an unidentified ailment. Players must explore the animal's environment, gather data, and ultimately craft an argument to explain why the animal is sick based upon the available evidence, while simultaneously learning about the monk seal's biome and habits. The process of collecting data and arranging statements to create an argument is more cognitively complex than simply recalling information or implementing a rote procedure.

Research has also indicated that how a student is able to approach a task can contribute to the cognitive demand of a task. Henningsen and Stein [74] found that classroom factors could shape cognitive demand, either maintaining or lowering the cognitive demand associated with an otherwise rigorous and challenging task. For example, allowing students to pursue a problem from multiple points of entry—rather than prescribing a specific entry point or strategy—allows for problems to be solved in a range of ways and using a range of resources. In terms of digital games, some have linear paths where a student must solve problems in succession and in certain ways to achieve the goals. Other games allow players to explore the environment and manipulate the environment's objects to become tools for achieving the game's objectives. In gaming, this has been partly explained as level of control, which is a heuristic of a model of enjoyment based on flow theory [75,76]. Literature on digital games has shown that the amount of control can influence a student's perception of learning and motivation [77–79]. These elements of control can take several forms, including student choice over (1) the features of their player avatar, (2) when and how they explore different parts of the game environment, and (3) the tools or methods needed to accomplish a goal. While student choice over features of an avatar may be more related to game aesthetics or cultural relevance than to learning complexity, choice over when and how to explore the game environment and how to accomplish a goal is more clearly implicated in learning complexity.

### 1.3.3. Cultural Relevance

The features and learning complexity of a game can influence a student's identity, sense of belonging, or mindset. These influences are driven by how a player perceives the messages provided by how one is represented within a game or expected to engage with the game's goals. Digital games contain cultural models, defined as the shared mental representation of cultural ideas, practices, and understandings of a group [80]. As with many domains of school and society, cultural models are inherent in the themes and narratives of digital games, and these cultural models have important implications for equity. As a diverse array of students enter any given educational space, they may have widely varying associations, reactions, and expectations to the cultural models in that space and in the curriculum [81]. Through gameplay, cultural models can affect a student's identity, defined as how the student sees themselves in relation to a particular community (e.g., a community of STEM scholars) and how the student believes they are perceived by others [82–84].

Students might learn more from games than merely academic content. Game themes, backstories, and narratives can portray racial/ethnic and gender stereotypes that can send negative messages about who the game is intended for and hinder student learning and in-game performance [51,85–89]. If students do not see their identities reflected in a game, this may alienate them from the game, reducing their learning; independent of this concern,

it may also convey harmful messages about belonging or non-belonging in the academic domain(s) or discipline(s) associated with the game. Teachers (and game designers) must consider how digital games could perpetuate negative messages and (re)produce inequity.

Lastly, a game's response to a player's mistakes and failures can promote different fixed or growth mindsets [90]. For example, if a student cannot progress through a game because of mistakes, it can lower engagement and send the implicit message that the student has a low ability regarding the content associated with the game. Yet, digital games have the potential to support productive struggle and risk-taking associated with promoting a growth mindset. According to Gee [9], digital games encourage failure as "learners can take risks in a space where real-world consequences are lowered" (p. 62).

The cultural relevance component of our framework asks teachers to consider how the theme, backstory, or details of a game might influence students' identity, sense of belonging, and mindset. Finally, we consider how game features, learning complexity, and cultural relevance in digital games fit into a broader context of technological pedagogical content knowledge for games (TPACK-G).

*1.4. Technological, Pedagogical, and Content Knowledge for Games (TPACK-G)*

1.4.1. Background of TPACK-G and Teacher Professional Development

Shulman [91] famously argued that teachers need multiple kinds of knowledge to be effective and transformative educators. These include content knowledge, curricular knowledge, and pedagogical content knowledge (that is, knowledge of how to teach specific content, beyond merely general knowledge about how to teach). Additionally, technology use has been empirically linked to student achievement in mathematics [92,93]. Since many teachers still struggle to effectively integrate technology in ways that positively impact learning [33,94,95], technological knowledge has become an additional body of knowledge that teachers need [96]. Furthermore, Niess [97] has argued that distinct technological pedagogical content knowledge, or TPACK, is needed by teachers to successfully use technology in support of student learning.

Digital games are one type of technology that can be used to support student learning [98]. Knowing how to 'choose and use' digital games to teach specific content is an important component of TPACK, a component sometimes referred to as TPACK-G [99]. Since our focus was on developing DGBL, we pulled from the Game Pedagogical Content Knowledge (GPCK) framework [99], which is an application of the technological component of the TPACK framework (see Figure 2).

Many teachers, even those with experience playing digital games, may not have personal experiences with digital game-based learning. Without PD and support, these teachers may struggle to implement DGBL effectively, or may use digital games for only a narrow range of purposes. Williamson [100] found that teachers tended to view games as tools for entertaining students or for reinforcing already taught concepts. Takeuchi and Vaala [19] surveyed 694 US K–8 teachers and 75 percent reported implementing games, but the vast majority used "drill and practice type" games that only offered practice of lower-level skills. Furthermore, the ways that teachers use digital games in their classrooms can also impact student learning outcomes [101,102]. STEM education has an urgent need for PD that can help teachers choose and implement games effectively, in ways that advance educational equity.

Until recently, research on TPACK addressed technology as a general intervention, with little focus on specific interventions [103]. Although there is substantial research on the development of teachers' TPACK, there are few studies on the development of teachers' TPACK-G, particularly with attention to equity. To date, the limited body of research on TPACK-G PD has largely focused on familiarizing teachers with digital games or providing them with specific types of technological skills. Some interventions exposed teachers to gameplay [50,104], while others studied teacher experiences or their shifts in beliefs after implementing a digital game [105–107]. Another study taught teachers to identify game requirements [108], and some studies engaged teachers in game design [23,109–111]. A few

studies have guided teachers through the full process of choosing a digital game, planning, implementing the game as part of instruction, and then reflecting on this implementation afterwards (cf. [109,112–114]). Yet, none of these studies also included explicit attention to issues of equity. To foster TPACK-G and promote teacher learning, we designed workshops using pedagogies of practice in professional education [115].

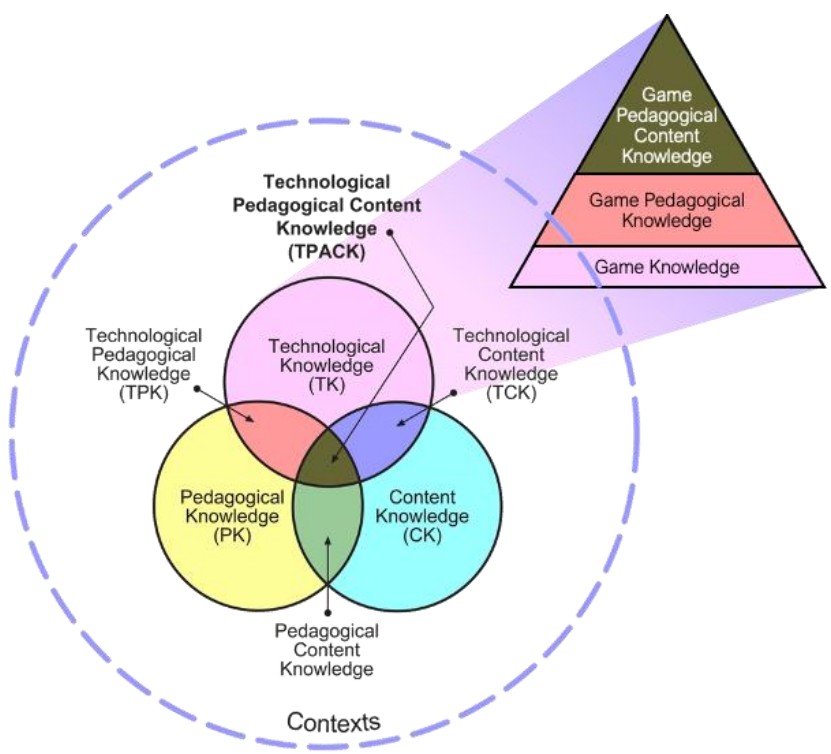

**Figure 2.** Technological, pedagogical, and content knowledge for games framework.

### 1.4.2. Pedagogies of Practice in Teacher Professional Education

Reviews of research on strategies to develop preservice teachers' digital literacy show that teachers need to have access to technology, received modeling of lessons, and collaboration across teacher educators and other teachers [116]. To prepare teachers for complex teaching methods requires implementation of deliberate pedagogies of teacher education [117]. Grossman and colleagues [115] proposed a framework for the teaching of practice. They identified three key concepts for the pedagogies of teacher education or practices: representations, decomposition, and approximations of practice. In the current study, we conducted workshops that involved representations of practice by allowing teachers to participate as learners while the authors modeled lessons using digital games in varied ways. To enact decomposition, the authors strategically engaged in meta-commentary about their pedagogical decision making and drew explicit connections between the Digital Game Evaluation Framework, learning goals, and implemented practices for the purposes of teaching and learning. These strategic "pauses" allowed opportunities for participants' questions where, as a whole group, we could anticipate issues and discuss practices that could address different contexts. Finally, the teacher participants engaged in approximations of practice by selecting and rehearsing digital games and planning game-based lessons. Teachers were given full autonomy in selecting a game and supported by the facilitators throughout the planning process. Then, the participants engaged in a cycle of offering and receiving feedback, revising and rehearsing their lessons, and offering and receiving further feedback. This process enabled the participants to prepare for implementing digital game-based learning in their own classrooms.

## 2. Materials and Methods

We engaged in design-based research [118–120] as we crafted and implemented this PD. Design-based research simultaneously pursues the goals of developing effective learning environments and using those environments as a basis for studying teaching and learning [118,121]. We set out with three main goals: generate theory about how teachers learn to use DGBL to equitably engage students in deep STEM learning; strengthen teacher self-efficacy in using games for deeper learning; and design a structure for professional learning that fosters these goals. With these goals in mind, we designed a PD to examine how these goals were being taken up by in-service teachers with their own goals of learning more about how technology can be used to develop students' STEM skills.

### 2.1. Participants and Context

The series of four 3-h workshops were designed to be practice-based, adaptive, responsive, and improvisational [115,122]. Online workshops were conducted by the authors using the teleconferencing platform Zoom. Participants were recruited through online social media posts as well as emails to professional networks and alumni of teacher education programs. Nine teachers ultimately applied to participate in the PD. To de-identify the relatively small sample of teachers, we present each racial/ethnic/national, gender, and teaching role demographic separately: four teachers were Latine, two were Mexican, three were white, and one was Filipinx American; six of these participants were female, and three were male; four taught multiple-subject elementary, one taught elementary mathematics, one taught secondary science, two taught secondary mathematics, and one had previously taught secondary mathematics and science but was teaching Spanish at the time of the study.

### 2.2. Data Collection and Analysis

2.2.1. Surveys

Before teachers started the PD, they were asked to take a pre-survey that consisted of a total of 28 items. The first 14 items were taken from the TPACK-G survey [99] and used a 7-point Likert scale to measure teachers' self-reported comfortableness and familiarity with digital games, subject matter knowledge in games, and capabilities for teaching with digital games. The next 14 items were adapted from Elementary STEM instruction component of the T-STEM survey [123]. The original items were not related to digital games and instead asked teachers how often they engage their students in particular types of STEM practices, such as making predictions, making observations, or recognizing patterns. We modified the question stem for these items to ask teachers how effective they believed digital games could be as tools for engaging students in these STEM practices. Answer choices used a 5-point Likert scale ranging from "not at all effective" to "extremely effective".

The same 28-item survey was administered at the end of the PD during the fourth and final workshop. Eight of the participating teachers completed pre-surveys and all nine completed post-surveys. Survey data were analyzed using a paired two-tailed *t*-test to ascertain whether there were statistically significant changes in teachers' TPACK-G and their beliefs about the affordances of DGBL.

2.2.2. Interviews

Of the nine participants, seven teachers consented to be interviewed. Each author conducted and recorded at least one of the seven interviews over Zoom. All interviews took place within approximately two weeks following the fourth and last meeting of the PD. Each interviewer followed a prewritten protocol that consisted of four constructs: (1) participant's understanding of the Digital Game Evaluation Framework, (2) TPACK-G for DGBL, (3) workshop experiences, and (4) suggestions for future DGBL PD. See Appendix A for the complete list of questions. Each interview lasted between 30 and 56 min, averaging 40 min and 24 s. In this manuscript, we analyze only the six interviews with teachers who were currently teaching STEM disciplines; we exclude the interview

with the teacher currently teaching Spanish as this teacher focused on the uses of games for language learning rather than for engaging in science or mathematics practices.

All interviews were audio-recorded and then transcribed using the automated transcription software otter.ai. The authors then cleaned each transcript by listening to its respective audio recording and correcting any inaccuracies in the transcriptions. All transcripts were then uploaded to the qualitative software Dedoose version 9.0. Interview transcripts were put through several cycles of coding. First, interviews were protocol-coded for the following preestablished categories [124]: teachers' beliefs about DGBL, perceptions of the Digital Game Evaluation Framework, activities that participants said contributed to their implementation of DGBL, and participant suggestions for improvement. Next, the authors engaged in pattern coding to identify themes in the data as they related to the four constructs of protocol [124]. Units of coding were full sentences or full paragraphs so that the theme was coded within the context of the participant's statement. Overlaps in codes were allowed. Then, all three authors met repeatedly to discuss emerging themes. Any disagreements were resolved through a process of negotiated agreement, which is where two or more coders discuss any coding differences and come to a consensus of the ultimate code [125,126]. Given we only had to code seven interviews and the focus of our constructs, there were very few disagreements, and all codes were resolved.

### 2.2.3. Structure of Professional Development

The authors designed the workshop series to develop teachers' TPACK-G and help teachers implement DGBL that promotes equity. The PD consisted of four three-hour workshops on Saturdays during Spring 2021. We chose to conduct the PD on Saturdays given the participants resided in different time zones and it was convenient for planning three hours for the participants' and authors' availabilities. The first three workshops were each a week apart. The final workshop took place three weeks later to allow participants time to implement planned DGBL lessons with their students and then meet afterwards to reflect on data collected during their lessons. A diagram of the workshop timeline is shown in Figure 3.

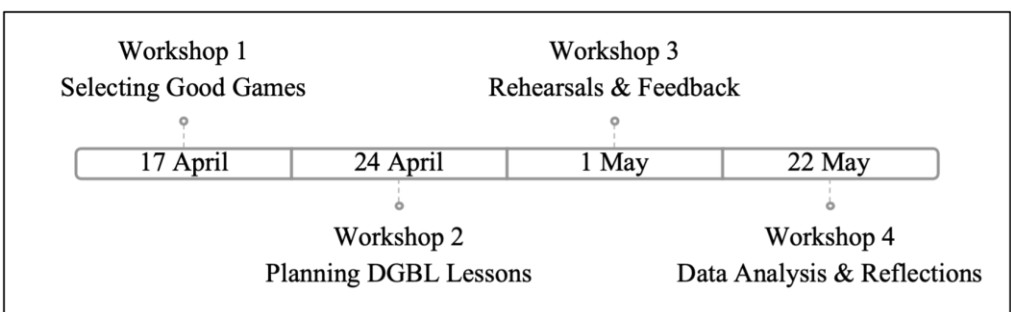

**Figure 3.** Timeline of workshops.

Workshop 1 focused on selecting and analyzing games using the Digital Game Evaluation Framework. It was designed to help teachers experience DGBL and identify the learning potential of digital games. The third author, Pope, began the workshop by briefly explaining the motivation, potential, and variety of digital games. She then guided participants through an experience of online digital gaming as a "student". At the time of the study, Pope was a postdoctoral fellow with Immersed Games as a researcher and Diversity, Equity, and Inclusion (DEI) consultant. As such, Pope granted participants access to the game Tyto Online to illustrate and critically analyze several key elements of the Digital Game Evaluation Framework. Each teacher participant was first prompted to create and customize a personal avatar. Options available for these avatars included a range of body types, hair colors, and skin colors; notably, game players are not prompted to choose a specific gender. Studies have shown that avatar representation is related to in-game performance and perceptions of STEM [127,128]. Participants then began playing the game, which

involved navigating a virtual world, gathering data, and crafting scientific arguments to help others solve problems, and assisting in refugee resettlement. Next, Pope discussed the Digital Game Evaluation Framework with participants, who used it to evaluate the game they had just played. Finally, participants were prompted to search out new games online and evaluate them using the same framework. During this time, participants were given a list of websites and apps of possible STEM games that the researchers had compiled over the years (e.g., NCTM Illuminations, Fog Stone Isle, DragonBox, BrainQuake, ScienceGameCenter, the Counting Kingdom, BrainPop, Embodied Games). They were encouraged to find a game that would be age-appropriate for their students, could be used to teach a STEM concept, and to play the games they found.

Play is crucial to learning, particularly for developing TPACK-G. Play expands how we engage with concepts and ideas and promotes transformational ways of thinking (Mishra et al., 2011). Starting with play has also been found to be an effective pathway to increasing teachers' game knowledge (GK) to then develop their game pedagogical knowledge (GPK) and game pedagogical content knowledge (GPCK) [99], all of which are components of TPACK-G.

Workshop 2 was designed to support teachers' DGBL lesson planning. Pope first modeled a DGBL lesson using The Factor Game (nctm.org). This is a two-player game (or one player versus the computer, or one teacher versus a whole class). Mathematical concepts of factors and multiples are fundamental parts of the gameplay. Pope then introduced the structure of a digital game-based lesson by sharing her lesson plan based on The Factor Game. Participants then deconstructed their experience for each lesson component. Grouped by grade levels, participants shared the games they found in their own online searches following the first workshop and began planning their own lessons using a lesson structure we recommended to them (see Figure 4).

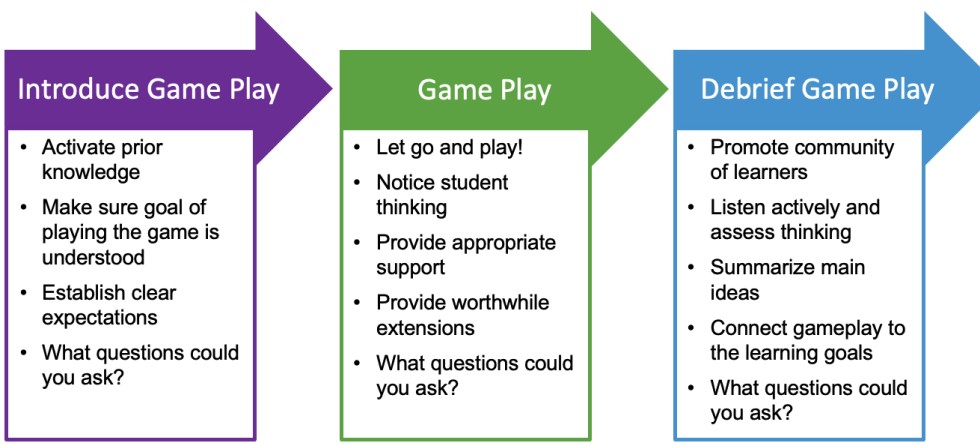

**Figure 4.** DGBL lesson structure.

Workshop 3 pushed participants to move beyond brainstorming and lesson-planning to rehearse a DGBL lesson for their own class. Participants were given time to create PowerPoint slides or other materials they might use while implementing DGBL during an upcoming lesson. As this PD was conducted while most schools were online due to the pandemic, the online format of the PD aligned well with the participants' teaching context. Participants were then sent into breakout rooms to share their slides with one or more colleagues and provide peer feedback. Each participant then took turns rehearsing the launch of their planned lesson, receiving feedback from colleagues, and revising their lesson plans. Rehearsals were designed as approximations of practice, a teacher education model intended to provide novice teachers with opportunities to practice in a safe environment [115]. At the end of this workshop, teachers were asked to consider how they would collect data and evidence of student learning to bring to the next workshop. The researchers gave examples of quantitative data and qualitative data that teachers could collect. For example, the presenters discussed sample lesson goals around mastery of a

standard and increasing student engagement. The data examples were a brief, standard-aligned multiple-choice test to obtain numerical data and completing an observation during lesson implementation to take notes about how students were engaging with the game and summaries of interactions with students to see how they liked the game play.

Workshop 4 was initially designed to promote critical reflection by directing teachers to engage in an evidence-based discussion about students' learning and experiences [129]. Being a design-based research PD, the goal of this workshop later changed goals to be responsive to the participants' needs. First, participants were prompted to make inferences based on data they collected from their students and what they could not infer but still wondered. The data collected by teachers varied. For example, two teachers had final scores of a game from each student; while another four had observation notes or 'exit tickets', which included content-based questions relating to the learning goals of the lesson; and three had no formal data. The workshop was not designed to conduct formal analyses of data but rather data were used to ground discussion in whether teachers could infer what students learned based on the data they had chosen to collect. We discussed the inferences that teachers made and discussed rationales for lessons with participants led by a researcher in two different breakout rooms. In the spirit of being responsive to the participants, some participants then asked the facilitators to engage a discussion regarding how to advocate for DGBL at their respective school sites. After offering some suggestions, participants talked about how they could bring this type of PD to their colleagues and how to convince administration to support it. The participants collaborated on a Google Doc to share form letters to parents, research they had found about the benefits of DGBL, how to collect useful data, who to partner with in promoting technologies, and more.

## 3. Results

### 3.1. Survey Results

#### 3.1.1. Teachers' TPACK-G

For quantitative analyses, Hedges' g was used to calculate effect sizes. The effects were large and statistically significant at the $p < 0.05$ level for every item on the TPACK-G instrument ($g > 0.8$ indicates a large effect size [130]). Most notably, teachers rated their ability to integrate digital games into teaching far higher on the post-survey (item 11; $g = 2.261$; $p < 0.001$) and rated their ability to identify instructional strategies for doing so far higher as well (survey item 10; $g = 2.023$; $p < 0.01$). See Table 1 for pre- and post-test means and Hedges' g for all TPACK-G survey items.

**Table 1.** TPACK-G survey results.

| Items | Mean | | $t$ | Hedge's $g$ | $p$ |
|---|---|---|---|---|---|
| | **Pre** | **Post** | | | |
| 1. I can learn digital games easily. | 5.38 | 6.63 | 3.035 | 0.960 | 0.019 * |
| 2. I can familiarize myself with the game interface. | 5.38 | 6.50 | 2.826 | 0.940 | 0.026 * |
| 3. I have the technical skills to play digital games effectively. | 5.38 | 6.75 | 2.762 | 1.230 | 0.028 * |
| 4. I know how to search for and download digital games. | 4.63 | 6.50 | 3.416 | 1.280 | 0.011 * |
| 5. I can identify the knowledge related to the subject matter in the digital games. | 4.25 | 6.63 | 4.771 | 2.136 | 0.002 ** |
| 6. I can tell whether the digital games represent the targeted subject matter knowledge. | 4.63 | 6.13 | 2.806 | 1.177 | 0.026 * |
| 7. I can identify whether the core concepts of the subject matter knowledge are displayed in the digital games. | 4.75 | 6.63 | 4.710 | 2.085 | 0.002 ** |
| 8. I can identify whether the subject matter knowledge is applied in digital games. | 4.5 | 6.50 | 5.292 | 2.160 | 0.001 ** |
| 9. I know how to use the characteristics of digital games to support teaching. | 4.63 | 6.63 | 5.292 | 1.886 | 0.001 ** |
| 10. I know the relevant instructional strategies of digital games. | 4.25 | 6.63 | 4.771 | 2.023 | 0.002 ** |

**Table 1.** *Cont.*

| Items | Mean | | $t$ | Hedge's $g$ | $p$ |
|---|---|---|---|---|---|
| | **Pre** | **Post** | | | |
| 11. I know how to integrate digital games into teaching. | 4.38 | 6.75 | 5.656 | 2.261 | <0.001 *** |
| 12. I can teach lessons that appropriately combine my teaching subject, digital games, and teaching approaches. | 4.38 | 6.38 | 3.347 | 1.475 | 0.012 * |
| 13. I can craft real world problems about the content knowledge and represent them through digital games to engage my students. | 4.13 | 6.13 | 6.110 | 1.604 | <0.001 *** |
| 14. I can select digital games to use in my classroom that enhance what I teach, how I teach and what students learn. | 4.38 | 6.50 | 5.338 | 2.129 | 0.001 ** |

* $p < 0.05$, ** $p < 0.01$, *** $p < 0.001$.

### 3.1.2. Perceptions of DGBL

For the modified T-STEM survey items, the effects were more heterogeneous. Teachers grew more likely to believe digital games could help students work in small groups (survey item 16; $g = 1.011$; $p < 0.05$), recognize patterns in data (survey item 20; $g = 1.242$; $p < 0.05$), complete activities with a real-world context (survey item 23; $g = 1.198$; $p < 0.05$), and reason abstractly (survey item 25; $g = 1.202$; $p < 0.05$). Notably, some comparisons exhibited large effect sizes of Hedges' $g > 0.8$ yet did not exhibit statistical significance. This may be due to not having a large enough sample [131]. For full results, see Table 2.

**Table 2.** Modified T-STEM survey results.

| Items<br>*How effective do you think digital games can be for engaging students in the following STEM practices?* | Mean | | $t$ | Hedge's $g$ | $p$ |
|---|---|---|---|---|---|
| | **Pre** | **Post** | | | |
| 15. Develop problem solving skills through investigations (e.g., scientific, design or theoretical investigations). | 3.88 | 4.5 | 1.488 | 0.703 | 0.180 |
| 16. Work in small groups. | 3.88 | 4.75 | 2.966 | 1.011 | 0.021 * |
| 17. Make predictions that can be tested. | 4.13 | 4.88 | 2.393 | 1.008 | 0.048 * |
| 18. Make careful observations or measurements. | 3.75 | 4.63 | 1.986 | 0.840 | 0.087 |
| 19. Use tools to gather data (e.g., calculators, computers, computer programs, scales, rulers, compasses, etc.). | 4.00 | 4.75 | 1.821 | 0.910 | 0.111 |
| 20. Recognize patterns in data. | 3.63 | 4.75 | 2.826 | 1.242 | 0.026 * |
| 21. Create reasonable explanations of results of an experiment or investigation. | 3.63 | 4.50 | 2.497 | 0.874 | 0.041 * |
| 22. Choose the most appropriate methods to express results (e.g., drawings, models, charts, graphs technical language, etc.). | 3.75 | 4.38 | 1.488 | 0.645 | 0.180 |
| 23. Complete activities with a real-world context. | 3.88 | 4.88 | 2.646 | 1.198 | 0.033 * |
| 24. Engage in content-driven dialogue. | 4.38 | 4.75 | 1.426 | 0.597 | 0.197 |
| 25. Reason abstractly. | 3.75 | 4.88 | 2.826 | 1.202 | 0.026 * |
| 26. Reason quantitatively. | 4.00 | 4.88 | 2.966 | 1.255 | 0.021 * |
| 27. Critique the reasoning of others. | 3.88 | 4.25 | 0.814 | 0.394 | 0.442 |
| 28. Learn about careers related to the instructional content. | 3.50 | 4.63 | 2.826 | 1.135 | 0.026 * |

* $p < 0.05$.

### 3.2. Interview Results

Overall, the qualitative data paralleled the quantitative data in showing teachers' expanding ideas about the uses of DGBL. In determining how teachers' perceptions of DGBL shifted, the interview analysis showed three main topics: benefits of DGBL, advocacy for DGBL, and utility of the Digital Game Evaluation Framework. Our data also told a more complex and ambivalent story about how teachers operationalized equity and "cultural relevance" in the context of DGBL. Themes in the interview findings are discussed below.

### 3.2.1. Benefits of DGBL

Teachers described changes in their beliefs about the possible learning benefits of using digital games. Five of the six STEM teacher interviewees discussed how their perception of the purpose of games shifted. One secondary mathematics teacher described this shift as follows:

> My mindset flipped from it's a review, to it can be used to teach just like I use exploratory lessons to teach concepts. If I can find the right game content, it can do the same thing . . . And it's also having that mindset flip to 'it's not just review; it can also be used for actual teaching' is the biggest impact it's had on me.

As described above, the teacher noted their beliefs about the use of digital games had shifted from being used for reviewing concepts students had already learned to the possibility of using digital games for learning new concepts. Within this same statement, though, the teacher also added that this was dependent on finding "the right game content."

### 3.2.2. Teacher Advocacy for DGBL

Teachers also described a desire to use digital games—in particular, high-quality digital games—more frequently than they had previously. As one elementary multiple-subject teacher said:

> After this workshop, I, I want to be very intentional about integrating games that are meaningful. And, you know, as I plan for next year, so I would definitely like to use them regularly.

This desire to plan for the use of digital games was not limited to participants' own classrooms, however. One elementary multiple-subject teacher described her active efforts to explain the potential benefits of DGBL to other teachers and administrators; during our final workshop, she shared (with fellow participants and with the authors) the text of an email she had sent to parents to explain her use of Minecraft Educational during English instruction. Several other participants remarked on the importance of such advocacy efforts; for this reason, we created a Google document in which all participants and researchers could collectively brainstorm ideas for how to advocate for DGBL resources and opportunities. These ideas included showing how certain games aligned with state standards; sharing lesson plans; gathering and analyzing student data from DGBL lessons; and utilizing COVID-19-related funding to support DGBL. In the post-workshop interviews, an elementary mathematics teacher also described his desire to organize a PD workshop of his own, to share information about DGBL with his colleagues.

### 3.2.3. Utility of the Digital Game Evaluation Framework

Teachers were asked how they used the Digital Game Evaluation Framework to select a digital game to be used with their students. Some teachers talked about expected aspects. For example, three teachers—one elementary mathematics and two elementary multiple-subject teachers—focused on the prompt about age-appropriateness of a game. Unexpectedly, though, one secondary science teacher mentioned that they applied the framework not to just select a game but also as a tool to analyze other activities they had completed in the past. This teacher even noted that they were contemplating how a digital game could influence the classroom environment:

> After seeing that evaluation, I went back and like looked into mine, right, the ones I've used before and the ones I selected. So, I was now careful, like okay, the images, right? Are they appropriate for the age? Are there adequate instructions? Are they really self-explanatory? Or do they need assistance? Is it fostering a good collaboration rather than a very competitive environment? What do I really like, competition or collaboration? So that, those are aspects that I have not looked into when selecting any type of games, not even digital or non-digital? So, it was very helpful.

Aside from obtaining an overall idea of how teachers used the framework, we also wanted to know more specifically how teachers were conceptualizing and operationalizing the component of cultural relevance.

3.2.4. Operationalizing Concepts of Equity and Cultural Relevance

Each teacher seemed to express multiple varying conceptions of what was meant by "cultural relevance," with important consequences. Notably, all six STEM teacher interviewees spoke at times about cultural relevance in terms of representation. Some explicitly referred to representations of marginalized racial, ethnic, or gender identities, while others discussed representation in vaguer terms.

Three of the interviewees described cultural relevance in terms of an affirmative need for representation. As one of the elementary multiple-subject teachers put it:

> It would be great if it reflects their background. You know, as a I was a student of color, I have students of color. So, it's, that is something that is very, very important, especially for the, you know, for my students to feel that they are represented and that they can relate.

In contrast, one secondary mathematics teacher pointed out that, even when games do reflect students' backgrounds, this may be harmful in certain situations. They discussed Tyto Online's narratives in which players are invited to assist with refugee resettlement; the mathematics teacher, who works with many refugee students, pointed out that "you could potentially tie into something traumatic to them if you weren't careful." This nuanced thinking could be valuable if it ultimately leads the teacher to use games in thoughtful ways that anticipate and proactively address such risks; at the same time, we are cognizant of the risk that some teachers may deliberately avoid using games that reflect students' backgrounds for such reasons.

At times, three of the interviewees characterized cultural relevance as an absence of bias in representation, using phrases such as "there was no implication...of discriminating" or "it's very neutral...not geared towards one population or another." Recognizing and avoiding overt bias or discrimination is certainly preferable to *not* recognizing such harms. However, searching for "neutrality" or the absence of bias implies that games can somehow be value- or culture-free, a problematic yet still-common assumption about STEM disciplines that has been extensively debunked [132,133]. One secondary mathematics teacher made this problem explicit when they described a particular game the following way:

> Pretty generic and pretty vanilla [*sic*]... You were in a spacesuit, you really couldn't see who you were, whatever you were, so you could just assume it was you.

This conception of cultural relevance implicitly presumes that all students (or at least all students included in the collective "you") are comparably likely to imagine themselves "in a spacesuit"—or, more evocatively, as "vanilla." This harmful colorblind ideology [134] can undermine efforts to promote educational equity. The same teacher de-emphasized the importance of racial representation when discussing Tyto Online and the significance of building a personal avatar within the game:

> It wasn't so much that it was skin colors, although there were skin colors, but there was, you know, hairstyles...if you were a funky person, or if you were nerdy person, or if you were, you know, all the different things I see in my classroom, you could come close to making something like that... I really could see that my kids would have fun. I mean, fun is not the end game of this. But if you have that fun, or if you get them engaged in that part of it, then it allows them to get to the learning part of it, which is what makes, you know, makes you move forward and walk forward.

Here, the teacher seems to suggest that representation is important insofar as it helps to advance academic achievement. This narrow conception of cultural relevance overlooks

other important goals of culturally relevant pedagogy, such as supporting students' cultural competence and critical consciousness [135].

In contrast, at least one elementary mathematics teacher seemed to be attending to both cultural competence and critical consciousness in their interview. This teacher argued that digital games need not merely passively reflect students' identities or actively impose identities upon them but can instead go further to serve as spaces for students to engage in active identity-building. This teacher described the importance of giving students "the ability to be able to choose who they want to play as, what they want to play as, whether it's a creature, or whether it's a human kind of being". They went on to discuss a preference for games that provide students opportunities for choice in self-representation:

> A boy character, or a girl character, and things like that, because I have students in sixth grade that are, you know, struggling with that. And it's, it's a new thing for them, that they're trying to, like, talk through and be able to solidify their own identity, or be able to just figure it out. And for them to just be like, be told, like, no, because you know, your name is this you have to be this...I didn't really want to, you know, put that on them. Instead...when students game they game, you know, and they just choose to be whoever they want...I really was looking forward...for them to have their own choice and their own voice in the matter.

While cultural competence can be defined in varied ways, one key interpretation is that students "become better persons in their own eyes, not just in the eyes of others" [136] (p. 19). In this excerpt, the teacher does not make assumptions about the elements of students' identities that will seem relevant or important to students; for example, they do not attempt to directly "match" gender or racial identities of game characters with their perceptions of students' gender or racial identities. Nor do they seek to find games that are purported to be free from "bias" in an abstracted, colorblind sense. Instead, they operationalize cultural relevance by attending to the variety of choices available to students for self-representation. Furthermore, they imply that such choices are not simply a tool for engaging students but important for the separate purpose of helping students "solidify their own identity or be able to just figure it out." Given the substantial research literature on identity development through digital gameplay [9,137,138], we argue that future research in DGBL should attend more closely to how teachers' implementation of digital games in their classrooms can expand, rather than constrain, students' opportunities for identity-building.

The same teacher also seemed interested in attending to issues of critical consciousness at a different point in their interview. The teacher described implementing a game in their own classroom in which students learned about fractions by cutting down trees and splitting logs to produce lumber for construction. The teacher explained that in this game:

> As you cut a log, you also had to plant a tree. So, it was really cool to like, show them like, deforestation doesn't have to be a thing...you can get the natural resources to build, but just make sure you replenish those natural resources. And I didn't have to say to that, I didn't have to tell them that strictly. I just had to let them play. And they kind of figured it out on their own. They were like, some were like, why do I plant a tree after I built something? And then it was my turn to be like, why do you think? And then they'd be like, because I just cut up a tree. Precisely the point. So that was really fun for them to be able to like, tie that to the current events, and the things that they're seeing in the news right now. So that they could, you know, be, ah, game changers in the future. And keep that in mind that it's not just about math, I don't want you here to come out just an excellent math student. I want you to come out of here being an excellent person.

Critical consciousness has been defined in various ways; Ladson-Billings argued that it refers to the ability and inclination to "critique the cultural norms, values, mores, and institutions that produce and maintain social inequities," [135] (p. 162). While this teacher did not explicitly discuss the inequities often produced and maintained by deforestation and depletion of natural resources, their comments do show they understand that digital

games embody specific values, and that the experience of playing a game can (under certain circumstances) prompt students to explore and interrogate these values. We argue that future research in DGBL should attend to how teachers identify the values embodied in a particular game, and how they might use such games to facilitate critical discussions of values in their classroom.

## 4. Discussion

This study contributes to the emerging research on TPACK-G [99,139] and equity in STEM education. Participants made sense of the Digital Game Evaluation Framework and applied it in their own classroom practice and in the process developed both stronger TPACK-G and expanded ideas about the variety of scientific and mathematical practices that DGBL can support.

In terms of advancing equity, our findings suggest that issues of representation were among the easiest aspects of cultural relevance for teachers to operationalize in DGBL. However, some teachers considered representation to be merely avoiding overt bias, while others talked about the importance of games affirmatively reflecting students' identities, and still others discussed games as spaces for active identity-building through students' choices for self-representation. Future PD and research should carefully attend to these patterns by working to confront and disrupt notions of colorblindness that may arise in discussions of DGBL. Indeed, interrogating the idea of "neutrality" or "colorblindness" in digital games may serve as a useful starting point for helping STEM educators interrogate these notions in the broader disciplines of science and mathematics.

Teachers also displayed considerable agency by asking for time to discuss advocacy for DGBL. In interviews, some teachers were critically evaluating their district curriculum and considering how it could affect student engagement. Others were considering how to lead a DGBL PD with their colleagues. This highlights the utility of practice-based, responsive PD for supporting teachers to continue developing equitable teaching practices in DGBL.

The PD also had some limitations that could be addressed in future iterations. Although we tried recruiting a larger pool of participants, we ultimately only had a total of nine teachers who all elected to participate. The fact that teachers self-selected and were a small number of participants restrict generalizations we can make about the PD's potential impact on a representative sample of STEM teachers. Further iterations of this work seek to implement this PD with larger groups of teachers but also wider roles in STEM education. In Workshop 4, the participants were asked to bring data from the lesson to analyze student learning or motivation. Not all participants collected data that could be formally analyzed, which meant the discussion was more grounded around what data teachers could have collected, and limitations to the data that can be captured from digital gameplay during a lesson. In terms of analyzing teachers' operationalization of equity and cultural relevance, the study did not conduct a pre-assessment nor analysis of how teachers were already conceptualizing these concepts nor how their own identities related to their operationalizations. This could be one explanation as to the wide variance in response from our participants. This needs to be further explored in future iterations of PD of this type to differentiate learning.

## 5. Conclusions

The PD was successful in increasing teachers' TPACK-G, as evidenced by the survey results. Interviews revealed that teachers saw the usefulness of implementing games for STEM learning, and teachers even discussed how to advocate for resources and time to discuss DGBL at their own school sites. When further asking teacher participants about the Digital Game Evaluation Framework, some teachers mentioned being more critical of the games they implemented in their classrooms. The one component of cultural relevance had the most variance in conceptualization and operationalization across all participants.

This study brings equity to the forefront of DGBL. Previous work around equity in DGBL is scarce but emerging [52,140]. Our findings suggest that teachers can make sense of and apply a framework such as the Digital Game Evaluation Framework for choosing and analyzing the utility of digital games in their instruction. Although the operationalizations of equity and cultural relevance varied, the framework showed some potential in evaluating games, as expressed by one participant who questioned whether games are reflective and beneficial for students' racial and gendered identity development. As STEM continues to suffer from more diverse participation, conversations around cultural relevance of the curriculum and tools used to teach STEM courses could be an important step in promoting STEM education. We encourage other authors and teacher educators to take up this work. At the same time, we also found that conceptions of cultural relevance were largely centered on issues of representation; these are of course significant, yet generations of research and advocacy teach us that representation is necessary but not sufficient. Future research should explore ways of expanding teachers' conceptions of cultural relevance and illuminating political clarity [141] in digital game-based learning. This work has the potential to address teachers' needs—and, even more importantly, it has the potential to help mitigate the ongoing (re)production of inequity in STEM education.

**Author Contributions:** Conceptualization, A.M.V.III, Q.C.S. and H.Y.P.; methodology, A.M.V.III, Q.C.S. and H.Y.P.; data collection, A.M.V.III, Q.C.S. and H.Y.P.; data analysis, A.M.V.III, Q.C.S. and H.Y.P.; writing—original draft preparation, A.M.V.III, Q.C.S. and H.Y.P.; writing—review and editing, A.M.V.III, Q.C.S. and H.Y.P.; funding acquisition, Q.C.S. All authors have read and agreed to the published version of the manuscript.

**Funding:** Participant stipends were supported by Southern Methodist University but did not receive any specific grant. Holly Pope was supported by the National Science Foundation (grant no. IIP-1853888).

**Institutional Review Board Statement:** The study was conducted in accordance with the Declaration of Helsinki and approved by the Institutional Review Board of Southern Methodist University (protocol code H21-013-SEDQ, approved 02/09/2021).

**Informed Consent Statement:** Informed consent was obtained from all subjects involved in the study.

**Data Availability Statement:** Data is unavailable due to privacy.

**Conflicts of Interest:** The authors declare no conflict of interest. The funders had no role in the design of the study; in the collection, analyses, or interpretation of data; in the writing of the manuscript; or in the decision to publish the results.

**Appendix A**

The interview protocol was organized by four main constructs.

1. Understanding of evaluation of a game framework
   a. How did you use the game evaluation framework to select a digital game to use with your students?
   b. How do you think this framework could be improved so that it's more useful for teachers?
   c. Now I want to focus on specific components of the framework.
      i. In recalling the component about cultural relevance, how do you think about or conceptualize that?
      ii. How did you recognize the presence or absence of cultural relevance in games?
      iii. When you think about whether content is integrated into the game mechanics, is it more helpful to think about it within the framework component of "learning complexity" or the component "game features"?
         1. PROBE: Why do you think that is more helpful when evaluating a digital game?

    2.    TPACK-G for DGBL

        a.    How has your capacity to use technology for teaching changed because of the workshop?

        b.    For what purposes do you feel the most comfortable using games and why?

            i.    PROBE: When would you use digital games throughout the year and for what purpose?

        c.    What do you think your greatest challenge will be with using digital games for learning in the classroom?

        d.    Explain how useful the lesson structure was for thinking about planning your digital game-based lesson. (Game Intro, Game Play, Game Debrief)

        e.    With this next question, I would like you to try and walk me through your process: if you were to design a new digital game-based lesson, what are some different elements you would now think about to implement a game for student learning?

        f.    Was there anything else from the workshops that helped you think differently about instruction or your teaching practices?

    3.    Workshop Experiences

        a.    Can you recall any moments during the workshops where you had a shift in thinking or an "aha" moment? If so, please describe them.

        b.    What can you tell me about collaborating with your peers during the workshops?

            i.    PROBE: How did you feel about interacting with new, unfamiliar colleagues over Zoom?

        c.    Thinking back on all the workshop sessions, is there anything else you would like to tell us about your experience?

    4.    Suggestions & Wrap-up (10 min)

        a.    What were the most helpful parts of the workshops?

        b.    Is there anything that was not helpful, and we could consider changing or trading out for a different activity?

        c.    Is there anything else you would like me or the team to know that I did not ask about today?

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
