# Peer review of "I DiG STEM: A Teacher Professional Development on Equitable Digital Game-Based Learning"

_education, doi:10.3390/educsci13090964_

Round 1
Reviewer 1 Report
The article explains the results of a Teachers' Professional Development program on Equitable Digital Game-Based Learning. The article is relevant to the Journal and is very well written. It is very comprehensive and easy to understand. The literature is well reviewed as well.
However, it could be improved by addressing the following clarifications:
1. Author3 developed digital game evaluation framework. While the components of the framework are explained in the article, it is not clear if it was proposed in this article or was it already published and re-used here? Kindly clarify.
2. In the 4th workshop, were the teachers asked to analyse the data that they collected? Were they given any tools to analyse their data or facilitate the process, considering that some of them might not have the skills needed for analysis? Kindly explain more about this process.
3. Please mention the interview process that was followed, and possibly indicate what kind of questions were asked from them.
4. Line 50-52 could use some re-structuring to make it easy to understand. Pose the research questions in their own sentence and briefly expand on them. It would be great to address each of these research questions in the discussion section separately with their implications mentioned.
Author Response
Response to Reviewer 1 Comments
Thank you for your comments. We, the authors feel your suggestions strengthened our argument and overall manuscript. All the changes suggested by Reviewer 1 were highlighted in yellow in the manuscript.
Point 1: Author3 developed digital game evaluation framework. While the components of the framework are explained in the article, it is not clear if it was proposed in this article or was it already published and re-used here? Kindly clarify.
Response 1: This framework was used in previous works. To clarify this, we add the following sentence after the mention of the framework: "This framework was developed and used in previous studies to analyze digital games used in classrooms (see )." For this review, we redacted those two citations to keep the manuscript blinded.
Point 2: In the 4th workshop, were the teachers asked to analyse the data that they collected? Were they given any tools to analyse their data or facilitate the process, considering that some of them might not have the skills needed for analysis? Kindly explain more about this process.
Response 2: This was a limitation of our PD that is good to highlight. We added some more context to the data collection in the paragraph about Workshop 4 and noted that the participants did not do a formal data analysis. Rather the data collected was used to ground the discussion in what the teachers could ascertain students learned based on the evidence they collected. The following includes the new sentences within that section:
"At the end of this workshop, teachers were asked to consider how they would collect data and evidence of student learning to bring to the next workshop. The researchers gave examples of quantitative data and qualitative data that teachers could collect. For example, the presenters discussed sample lesson goals around mastery of a standard and increasing student engagement. The data examples were a brief, standard-aligned multiple-choice test to obtain numerical data and doing an observation during lesson implementation to take notes about how students were engaging with the game and summaries of interactions with students to see how they liked the game play.
Workshop 4 was initially designed to promote critical reflection by directing teachers to engage in an evidence-based discussion about students’ learning and experiences [122]. Being a design-based research PD, the goal of this workshop later changed goals to be responsive to the participants needs. First, participants were prompted to make inferences based on data they collected from their students and what they could not infer but still wondered. The data collected by teachers varied. For example, two teachers had final scores of a game from each student; while another four had observations notes or ‘exit tickets,’ which included content-based questions relating to the learning goals of the lesson; and three had no forma data. The workshop was not designed to do formal analyses of data, but rather data was used to ground discussion in whether teachers could infer what students learned based on the data they had chosen to collect. We discussed the inferences that teachers made and discussed rationales for lessons with participants led by a researcher in two different breakout rooms. In the spirit of being responsive to the participants, some participants then asked the facilitators to open a discussion of how to advocate for DGBL at their respective school sites. After offering some suggestions, participants talked about how they could bring this type of PD to their colleagues and how to get administration to support it. The participants collaborated on a Google Doc to share form letters to parents, research they had found about the benefits of DGBL, how to collect useful data, who to partner with in promoting technologies, and more."
This meant that we also included more in the discussion to explain that the PD was limited to discussing data rather than doing a more methodologically sound analysis of the teachers’ data.
Point 3: Please mention the interview process that was followed, and possibly indicate what kind of questions were asked from them.
Response 3: Under section 2.2.2. Interviews, we added the following lines: "Each interviewer followed a prewritten protocol that consisted of four constructs: (1) Participant’s understanding of the Digital Game Evaluation Framework, (2) TPACK-G for DGBL, (3) workshop experiences, and (4) suggestions for future DGBL PD. See Appendix A for the complete list of questions."
The interview protocol was added as Appendix A.
Point 4: Line 50-52 could use some re-structuring to make it easy to understand. Pose the research questions in their own sentence and briefly expand on them. It would be great to address each of these research questions in the discussion section separately with their implications mentioned.
Response 4: The research questions were written as explicit questions:
"This study was guided by the following research questions: (1) How did the PD increase teachers’ TPACK-G? (2) How did teachers’ perceptions of DGBL shift? And (3) How did teachers operationalized concepts of equity and cultural relevance?"
The structure of the results also changed to following this format. 3.1 is now titled, "Survey Results Showed Increases in Teachers' TPACK-G," and the section 3.2 Interview Results explicit shares that there were three main topics that were found in answering the shift in teachers' perceptions of DGBL before getting to the last section in the results about how teachers operationalized concepts of equity and cultural relevance.
Reviewer 2 Report
This is an interesting paper, it discusses teacher professional development (PD) on Equitable digital GBL. The purpose of the study according to the authors is to inquiry whether the PD would increase teachers’ TPACK-G, how teachers’ perceptions of DGBL may shift, and how teachers operationalized concepts of equity and cultural relevance. However, it does not discuss what it means by the word “equity”. What is equity? What do you mean by equitable? How do you define equity? Terminologies need redefining.
The study then moves on to provide an overview of a framework for evaluating digital games and how K-12 STEM teachers (n=9) made sense of the framework and exhibited shifts in TPACK-G. But this needs to be clearer. Why did authors not mention that the study used mixed methods from the beginning. The paper is confusing in this sense. The paper then moves on to discuss findings saying “..teachers can make sense of and apply an equity-centered framework for choosing and using digital games in their instruction, but how? Could you provide examples? Could you make this obvious?
The abstract needs rewriting completely. The authors suggest that the sample is 9 teachers. How is this credible? When in fact later on when you read on you discover that the study used mixed research methods, pre- surveys and post surveys and interviews. However, the author/s never mention this from the beginning.
Discussion is too short and requires more clarity about how themes in results relate and answer the research questions.
Conclusion is vague, it should be made more clearer and realistic in terms of results and what the limitations are. Instead of saying it was all 'successful'. The authors should discuss it and made this obvious.
The paper is readable but the grammar needs attention throughout e.g.
“type games that only offered practice” type of games that …
The sub heading numbering is wrong and needs revisiting they put 1.2 in the heading then subheading 1.3.1 straight after this.
Author Response
Response to Reviewer 2 Comments
Thank you for your comments. We, the authors feel your suggestions strengthened our argument and overall manuscript. All the changes suggested by Reviewer 1 were highlighted in green in the manuscript.
Point 1: This is an interesting paper, it discusses teacher professional development (PD) on Equitable digital GBL. The purpose of the study according to the authors is to inquiry whether the PD would increase teachers’ TPACK-G, how teachers’ perceptions of DGBL may shift, and how teachers operationalized concepts of equity and cultural relevance. However, it does not discuss what it means by the word “equity”. What is equity? What do you mean by equitable? How do you define equity? Terminologies need redefining.
Response 1: We included a new section (1.2) in the introduction to discuss equity in terms of digital game-based learning. See the following:
1.2. The Potential of DGBL to Address Equity and the “Digital Divide”
The “digital divide” refers to the notable differences in access to digital technologies between prvileged and historically marginalized communities [36]. This divide is further exacerbated by the quality of digital technologies different populations of students have access to for deep learning. This PD was designed to help close this divide by introducing, evaluating, and implementing digital games as tools for equitable learning.
Equity can be characterized by the following three components: fair distribution of opportunities to learn, equal opportunites to participate, and equal educational outcomes. These components affirm DGBL as an equitable pedagogical approach and were utilized in the design and implementation of the I DiG STEM PD.
The fair distribution of opportunities to learn has long been framed by socological approaches generally around the resources available for a student to achieve (e.g., textbooks, instructional strategies, materials, teacher’s background, class size, time, quality) [37–39]. Tate [40] extends this definition to emphasize the importance technology for learning science, highlighting projects and technology partnerships that offer students equipment for hands-on inquiry and improved STEM instruction. The National Council of Teachers of Mathematics gave a position on the equitable integration of technology to “be used to develop and deepen learner understanding, stimulate interest in mathematics being learned, and increase mathematical proficiency” [42]. Incorporating different technologies can expand ways that students learn and participate in STEM. Yet, research shows that gaining access to technological tools is not enough. Teachers’ qualifications, knowledge, beliefs, and experience with digital technologies are important components of equity with respect to implementing learning resources [43].
Teachers’ abilitiies to implement digital technologies has implications for how students have opportunities to participate in learning. Studies have highlighted that although students have more access now, there is still a “usage gap,”[44] which emphasizes how technology is simplify a substitution or augementation to traditional teaching methods or used to modify or redefining learning [37,45–47]. For example, in a case study with four middle schools teachers of English learners, Siebert et al. [48] found that technology was used as task substitution (e.g., organizational tool for online quizzes, ReadWorks to determine a student’s Lexile level) or to enhance instructional materials (e.g., instruction with digital worksheets, Nearpod Educator, or PowerPoint) rather than purposeful student-centered engagement or deep learning.
Lastly, equity must also address student outcomes. Student outcomes can be describe the learning experience or educational achievements. Studies have shown the potential that DGBL has on improving student outcomes. For example, Kebritchi et al. [49] found significant improvements in motivation and mathematics achievement for a diverse group of students from an urban school after playing a computer game. Further, McLaren et al. [50] found that students with low prior knowledge actually showed the highest gains in mathematics learning from game play.
Thus, by analyzing DGBL with these components equity, it shows potential in addressing the digital divide and promoting equitable STEM learning.
Point 2: The study then moves on to provide an overview of a framework for evaluating digital games and how K-12 STEM teachers (n=9) made sense of the framework and exhibited shifts in TPACK-G. But this needs to be clearer. Why did authors not mention that the study used mixed methods from the beginning. The paper is confusing in this sense.
Response 2: The study reports on a pilot study with 9 participant teachers. We are unsure what the reviewer would require for the study to be more “credible.” We are not aiming to make any generalizations about such a professional development but rather reporting on the results of this pilot. We did include in the following paragraph after a re-write of the research questions that was suggested by Reviewer 1:
“This mixed methods pilot study used pre- and post-surveys and interviews to investigate shifts in teachers’ (n=9) TPACK-G, perceptions of DGBL, and operationalizations of cultural relevance. Survey results showed increases in teachers’ TPACK-G, and corroboration between surveys and interviews showed teachers’ expanded perceptions about the range of applications of digital games in STEM learning. However, interviews revealed that teachers’ conceptualizations of cultural relevance varied considerably. Teachers emphasized the importance of representation, but some emphasized a need to affirm representations, whereas others conceptualized it as an absence of bias.”
This is also addressed in Point 4, where we re-wrote the abstract to indicate it was mixed methods. We also included a similar statement after the research questions.
Point 3: The paper then moves on to discuss findings saying “..teachers can make sense of and apply an equity-centered framework for choosing and using digital games in their instruction, but how? Could you provide examples? Could you make this obvious?
Response 3: This was stated in the conclusion, which has been extended to give more examples of this and also address Point 6.
Point 4: The abstract needs rewriting completely. The authors suggest that the sample is 9 teachers. How is this credible? When in fact later on when you read on you discover that the study used mixed research methods, pre- surveys and post surveys and interviews. However, the author/s never mention this from the beginning.
Response 4: The abstract did state that the evaluation was mixed methods, but we went ahead and separated the sentence to be more direct and also indicate the use of surveys and interviews. Abstract was re-written to state the following:
“Digital game-based learning (DGBL) has potential to promote equity in K-12 STEM education. However, few teachers have expertise in DBGL, and few professional development models exist to support teachers in both acquiring this expertise and advancing equity. To support the development of such models, we conducted a professional development to explore teacher acquisition of technological, pedagogical, and content knowledge for games (TPACK-G) during a DGBL workshop series informed by culturally relevant pedagogy. This mixed methods pilot study used pre- and post-surveys and interviews to investigate shifts in teachers’ (n=9) TPACK-G, perceptions of DGBL, and operationalizations of equity and cultural relevance. Survey findings showed increases in teachers’ TPACK-G, and corroboration between surveys and interviews showed teachers’ expanded ideas about the range of applications of digital games in STEM education. However, interviews revealed that teachers’ conceptualizations of equity and cultural relevance varied considerably.”
Point 5: Discussion is too short and requires more clarity about how themes in results relate and answer the research questions.
Response 5: Highlighted portion of the discussion shows what was added to offer more clarity of the themes of the results and how they answered the research questions. Mainly, the discussion was written to parallel and expand on the research questions and results.
Point 6: Conclusion is vague, it should be made more clearer and realistic in terms of results and what the limitations are. Instead of saying it was all 'successful'. The authors should discuss it and made this obvious.
Response 6: We made the conclusion more specific to the study to make explicit the “successful” portions of the PD and those that are left unanswered.
Point 7: Comments on the Quality of English Language
The paper is readable but the grammar needs attention throughout e.g.
“type games that only offered practice” type of games that …
Response 7: This was worded correctly in the context of the sentence since “drill and practice” is the type of games we were discussing. We put the quotations around “drill and practice type” to hopefully make this clearer. We were unsure what other grammar the reviewer was referring to, but we put the paper through a grammer check to mitigate any grammar errors.
Point 8: The sub heading numbering is wrong and needs revisiting they put 1.2 in the heading then subheading 1.3.1 straight after this.
Response 8: Headings and subheading numbering was corrected. Because of added section on equity, The subheading is now numbered 1.4, and the two subsections are numbered 1.4.1 and 1.4.2.
Reviewer 3 Report
Dear authors, The paper has been well structured meanwhile following point address and justify in the research paper and resubmit for further review.
1. How was the sample size for the study determined?
2. Mention the details concerning the demographic composition of the participating teachers?
3. Specify the selection process for the specific digital games used for the workshop series?
4. Can you provide additional detail on how the automated transcription software was utilized to achieve reliable and accurate transcriptions?
5. Could you explain more about how teachers' perceptions of the digital game evaluation framework were measured and assessed?
6. How did you accommodate the teachers who might have had previous experiences or higher comfort levels with digital games in your study?
7. Add some more specific examples of the "work in small groups", "recognize patterns in data", and "complete activities with a real-world context", which were indicated as aspects teachers believed digital games could help?
8. Mention the rationale behind conducting the professional development (PD) workshops on Saturdays particularly? Could this have influenced the participation or results?
9. How did you ensure the confidentiality and anonymity of the participants during the interview process?
10. More detail on the “equity-centered framework” referenced in the study could be beneficial - what does it consist of and why was it chosen?
11. Could you provide additional data or specific examples to elaborate on how teachers were engaging in advocacy work to support DGBL?
12. Can you provide specifics on how "negotiated agreement" was reached when resolving disagreements among authors?
13. Mention if any measures were taken to control potential bias as all authors of the study were directly involved in interviewing the participants?
Average
Author Response
Response to Reviewer 3 Comments
Thank you for your comments. We, the authors feel your suggestions strengthened our argument and overall manuscript. All the changes suggested by Reviewer 1 were highlighted in cyan in the manuscript.
Point 1: How was the sample size for the study determined?
Response 1: The number of teachers were not per-determined. We sent out several emails through our respective networks, and these were the teachers we were able to recruit for the PD. This is why we stated that after recruting through social media, we ultimately got nine participants. We re-phrased to make this clearer: “Participants were recruited through online social media posts as well as emails to pro-fessional networks and alumni of teacher education programs. Nine teachers ultimately applied to participate in the PD.” We further address this in the discussion as a limitation. We added the following:
“Although we tried recruiting a larger pool of participants, we ultimately only had a total of nine teacher who all elected to participate. The fact that teachers self-selected, and the small number restricts generalizations we can make about the PD’s potential impact on a representative sample of STEM teachers. Further iterations of this work seek to implement this PD with larger groups of teachers but also wider roles in STEM education.”
Point 2: Mention the details concerning the demographic composition of the participating teachers?
Response 2: We included the gender and racial/ethnic/national demographics of our participants under section 2.1. To de-identify the teachers, each demographic is reported separately:
“To de-identify the relatively small sample of teachers, we present each racial/ethnic/national, gender, and teaching role demographic separately: four teachers were Latine, two were Mexican, three were white, and one was Filipinx; six of these participants were female, and three were male; four taught multiple-subject elementary, one taught elementary mathematics, one taught secondary science, two taught secondary mathematics, and one had previously taught secondary mathematics and science but was teaching Spanish at the time of the study.”
Point 3: Specify the selection process for the specific digital games used for the workshop series?
Reponse 3: We included more details about how TytoOnline was chosen and what other game resourses were shared during Workshop 1, which is under section 2.2.3. Structure of Professional Development:
“At the time of the study, Author3 was a postdoctoral fellow with Immersed Games as a researcher and Diversity, Equity, and Inclusion (DEI) consultant. As such, Author3 granted participants access to the game Tyto Online to illustrate and critically analyze several key elements using the Digital Game Evaluation Framework.”
Within this same paragraph about Workshop 1, we conclude with more details on the resources teachers were given to find and evaluate games:
“Finally, participants were prompted to search out new games online and evaluate them using the same framework. During this time, participants were given a list of websites and apps of possible STEM games that the researchers had compiled over the years (e.g., NCTM Illuminations, Fog Stone Isle, DragonBox, BrainQuake, ScienceGameCenter, the Counting Kingdom, BrainPop, Embodied Games). They were encouraged to find a game that would be age-appropriate for their students, could be used to teach a STEM concept, and to play the games they found.”
Point 4: Can you provide additional detail on how the automated transcription software was utilized to achieve reliable and accurate transcriptions?
Response 4: We used otter.ai. This was added to the methods. We then cleaned the transcriptions by listening to the audio and fixing any inaccurate transcriptions by hand. We went ahead and rephrased this part of the methods:
“All interviews were audio recorded and then transcribed using the automated transcription software otter.ai. The authors then cleaned each transcript by listening to its respective audio recording and correcting any inaccuracies in the transcriptions. All transcripts were then uploaded to the qualitative software Dedoose.”
Point 5: Could you explain more about how teachers' perceptions of the digital game evaluation framework were measured and assessed?
Response 5: Reviewer 2 had a similar question, and we addressed some of this in the methods and the results. We interviewed teachers specifically on their perceptions of the framework. The interview protocol is now included as Appendix A.
Point 6: How did you accommodate the teachers who might have had previous experiences or higher comfort levels with digital games in your study?
Response 6: This was a great question because it ended up not being an issue. We gave each participant choice of what game to implement, and even those teachers that self-identified as “gamers” found the experience beneficial because of the focus on implementing games for STEM learning. I could not find an appropriate place to indicate this in the manuscript as we did not officially assess the teachers’ experiences and comfort levels with games. The PD allowed for teachers to choose their own games to evaluate and implement. We included this comment in the Workshop 1 context and hope it is enough to explain how the PD was differentiated.
Point 7: Add some more specific examples of the "work in small groups", "recognize patterns in data", and "complete activities with a real-world context", which were indicated as aspects teachers believed digital games could help?
Response 7: These phrases were adapted items from an existing, validated survey intended to measure beliefs of DGBL. We did not inquire into each measure. Rather, we report on the fact that teachers felt that these measures said teachers felt digital games were more effective in engaging students in those STEM practices.
Point 8: Mention the rationale behind conducting the professional development (PD) workshops on Saturdays particularly? Could this have influenced the participation or results?
Response 8: We included the following text: “We chose to conduct the PD on Saturdays given the participants resided in different time zones and it was convenient for planning three hours for teacher participants’ and the authors’ availabilities during the academic school year.” We are all teachers and educators. So, Saturdays during the school year made the most practical sense.
Point 9: How did you ensure the confidentiality and anonymity of the participants during the interview process?
Response 9: We are not sure how to address this point explicitly in the manuscript, but all interviews were done over a private Zoom room in our respective homes and offices. All three authors knew the names and locations of each participant, but none of that is shared with anyone else. I do not normally see a statement of confidentiality or anonymity for qualitative studies. Is there something that the reviewer suggests?
Point 10: More detail on the “equity-centered framework” referenced in the study could be beneficial - what does it consist of and why was it chosen?
Response 10: We believe the reviewer is referring to a phrase used in the conclusion, where we stated that our finding suggest that teacher can make sense of and apply an equity-centered framework. We changed this to explicitly state the Digital Game Evaluation Framework as a tool for choosing and analyzing the utility of digital games in their instruction. We added more in the conclusion to include that the framework was potentially helpful in considering how some games could reflect students’ racial and gendered identities as highlighted by one participant.
Point 11: Could you provide additional data or specific examples to elaborate on how teachers were engaging in advocacy work to support DGBL?
Response 11: We included some more information about what teachers discussed and recorded in the Google Doc they created.
Point 12: Can you provide specifics on how "negotiated agreement" was reached when resolving disagreements among authors?
Response 12: This was explained further in the methods:
“Next, the authors engaged in pattern coding to identify themes in the data as they related to the four constructs of protocol [132]. Units of coding were full sentences or full paragraphs so that the theme was coded within the context of the participant’s statement. Overlaps in codes were allowed. Then, all three authors met repeatedly to discuss emerging themes. Any disagreements were resolved through a process of negotiated agreement, which is where two or more coders discuss any coding differences and come to a consensus of the ultimate code [133,134]. Given we only had to code seven interviews and the focus of our constructs, there were very few disagreements, and all codes were resolved.”
We also included another citation that explains more about how negotiated agreement works for interview coding: Campbell, J.L.; Quincy, C.; Osserman, J.; Pedersen, O.K. Coding In-Depth Semistructured Interviews: Problems of Unitization and Intercoder Reliability and Agreement. Sociol. Methods Res. 2013, 42, 294–320, doi:10.1177/0049124113500475.
Point 13: Mention if any measures were taken to control potential bias as all authors of the study were directly involved in interviewing the participants?
Response 13: This was addressed by including the interview protocol. Each author followed the questions in the protocol.
Round 2
Reviewer 1 Report
The article is good to submit. there are a few minor grammatical errors in the newly added text that could be fixed.
The article is good to submit. there are a few minor grammatical errors in the newly added text that could be fixed.
Reviewer 2 Report
Thank you for improving the paper. it reads nicely and it is much better.
Good luck.
Thank you for improving the paper. it reads nicely and it is much better.
Good luck.